# Success-efficient/failure-safe strategy for hierarchical reinforcement motor learning

Jan Babič[1,*], Tjasa Kunavar[1,2], Erhan Oztop[3,4], Mitsuo Kawato[5]

1 Laboratory for Neuromechanics and Biorobotics, Department of Automatics, Biocybernetics and Robotics, Jožef Stefan Institute, Ljubljana, Slovenia, 2 Jožef Stefan International Postgraduate School, Ljubljana, Slovenia, 3 Ozyegin University, Istanbul, Turkiye, 4 Osaka University, Osaka, Japan, 5 Brain Information Communication Research Laboratory Group, Advanced Telecommunications Research Institute International, Kyoto, Japan

❂ These authors contributed equally to this work and share first authorship.
* jan.babic@ijs.si

## Abstract

Our study explores how ecological aspects of motor learning enhance survival by improving movement efficiency and mitigating injury risks during task failures. Traditional motor control theories mainly address isolated body movements and often overlook these ecological factors. We introduce a novel computational motor control approach, incorporating ecological fitness and a strategy that alternates between success-driven movement efficiency and failure-driven safety, akin to win-stay/lose-shift tactics. In our experiments, participants performed squat-to-stand movements under novel force perturbations. They adapted effectively through various adaptive motor control mechanisms to avoid falls, reducing failure rates rapidly. The results indicate a high-level ecological controller in human motor learning that switches objectives between safety and movement efficiency, depending on failure or success. This approach is supported by policy learning, internal model adaptation, and adaptive feedback control. Our findings offer a comprehensive perspective on human motor control, integrating risk management in a hierarchical reinforcement learning framework for real-world environments.

## Author summary

Understanding the mechanisms underlying motor learning and control is crucial for advancing both neuroscience and robotics. This study introduces a novel framework that integrates ecological aspects into motor learning, emphasizing the dual objectives of movement efficiency and safety. By employing a hierarchical reinforcement learning model, we demonstrate how human sensorimotor systems adapt to perturbations through a dynamic balance between risk management and energy optimization. Our findings reveal that motor adaptation is not only a process of refining movement accuracy, but also a critical function

**Data availability statement:** All data underlying the findings of this study are available at Figshare: https://doi.org/10.6084/m9.figshare.28651805.v1. The model and simulation code is freely available at GitHub: https://github.com/erhan2umi/EcologicalHRL under Creative Commons Zero v1.0 Universal licence.

**Funding:** J.B. and T.K. were supported in part by the European Union's Horizon 2020 through the projects AnDy (contract nr. 731540) and SPEXOR (contract nr. 687662), by the European Union's Horizon Europe through the project SWAG (contract nr. 101120408), and by the Slovenian Research Agency (research core funding nr. P2-0076). E.O. was supported in part by JSPS KAKENHI (grant nr. JP23K24926). M.K. was supported by Grants-in-Aid for Transformative Research Areas (22H05156), Japan Agency for Medical Research and Development (AMED), Grant Number JP21dm0307008 and JP24wm0625502, and Innovative Science and Technology Initiative for Security Grant Number JPJ004596, Acquisition, Technology & Logistics Agency (ATLA), Japan. The funders had no role in study design, data collection and analysis, decision to publish, or preparation of the manuscript.

**Competing interests:** The authors have declared that no competing interests exist.

for survival in real-world environments. This research provides new insights into the complexity of human motor control. Our findings have potential applications in developing more adaptive and resilient robotic systems, as well as in clinical interventions for motor impairments.

## Introduction

Evolution has endowed us with motor learning abilities to boost our survival odds by enhancing competitive success [1]. Ecological fitness refers to the specific traits of an organism and abilities that aid survival in its environment [2,3]. The sensorimotor system is crucial for ecological fitness, enabling adaptive movement control vital for survival [4]. Therefore, understanding sensorimotor adaptation and motor control optimality must consider ecological fitness. Overlooking this aspect leads to incomplete sensorimotor models that do not fully capture the brain-body-environment interaction. The adaptive capacity of the sensorimotor system, in addition to the evolutionary optimization that has occurred, is subject to the continued refinement of movement policy throughout an organism's life [5]. Thus, a thorough grasp of the sensorimotor system demands a multidisciplinary approach that embraces ecological fitness and integrates recent neuroscience and evolutionary biology insights [6].

Recent advancements in behavioural research have elucidated several distinct learning processes inherent in motor skill acquisition, involving a variety of neuronal substrates and computational processes [5,7–11]. Among these, the basal ganglia are crucial for reinforcement learning [12–14], which facilitates the refinement and acquisition of novel motor skills by providing feedback that rewards successful movements and discourages ineffective ones. Motor behaviour can thus be shaped through experience and practice. This is carried out in interaction with internal models of motor control, by updating predictions and adjustments based on feedback to enhance the ability of the brain to anticipate and execute precise motor actions [15,16].

Internal models are neural representations that predict the sensory consequences of movements and generate feed-forward motor commands, enabling precise and efficient motor control [15–18]. These models are believed to reside partly in the cerebellum and are essential for human motor control. Past research has emphasized the role of the cerebellum in updating motor predictions based on sensory feedback to refine movement execution [5,6]. The brain also adjusts feedback control gains in response to movement errors, dynamically balancing predictive control with real-time corrections [19,20]. Optimal feedback control theory explains how the brain computes motor commands in real time by balancing movement accuracy against energy costs [21,22], and empirical studies confirm that the human sensorimotor system operates according to these principles [1,6].

Beyond optimizing movement efficiency, motor control must integrate environmental risk assessment when planning and executing movements [23]. Research in neuroeconomics suggests that the nervous system incorporates uncertainty into

decision-making, dynamically adjusting motor strategies to balance performance with safety [24]. This framework provides insights into the neural and computational mechanisms underlying motor planning in uncertain environments [25]. Risk-sensitive motor adaptation is particularly evident in gait and postural control, where stability maintenance is crucial to prevent falls. Studies have shown that humans adjust step width, base of support, and foot placement in response to perceived instability [26]. Similarly, research has demonstrated that metabolic cost interacts with risk perception in shaping movement strategies, emphasizing the continuous trade-off between efficiency and stability [27].

We propose that the optimality concept in computational motor control needs a redefinition from an evolutionary perspective, where ecological fitness is central in guiding efficient movement generation. This approach would allow traditional cost functions, which relate to efficient movement execution, to be balanced alongside injury risk reduction. By movement efficiency we refer to the optimal use of muscles to perform movements with maximal performance and minimal energy expenditure. We hypothesize that the human brain employs a *success-efficient/failure-safe* strategy, wherein success reinforces energy-efficient movement execution, whereas failure enhances risk aversion and prioritizes safety. This approach suggests a hierarchical reinforcement learning mechanism that dynamically adjusts motor learning objectives based on past experiences.

To test our hypotheses, we developed a unique experimental approach that overcomes the limitations of prior studies by incorporating ecological fitness, specifically exposing the sensorimotor system to potential failure and injury risk. In our study, participants engaged in unrestrained whole-body motions, performing squat-to-stand movements under significant force perturbations. Unlike traditional paradigms that focus on isolated limb movements, this experiment examines whole-body motion, highlighting the ecological constraints that shape motor learning while introducing real-world risks such as falling and injury, which directly affect ecological fitness of participants. Moreover, the high energy demands of squat-to-stand movements necessitate a sophisticated neural strategy that balances energy optimization with stability maintenance, further emphasizing the brain's adaptive control mechanisms. By adopting this ecologically valid framework for studying sensorimotor control, we seek to integrate human motor control into a broader hierarchical reinforcement learning model, where ecological fitness influences adaptation and learning throughout the life of an organism.

## Results

A group of participants performed a series of whole-body stand-up motions (trials) while their centre of mass (COM) was systematically perturbed in the backward direction. On average the movement time was (Mean±SEM) 1.16±0.067 s. Perturbations were accomplished by a force-controlled pulling mechanism (Fig 1A) with a magnitude proportional to the vertical velocity of the COM. The protocol involved four types of trials; baseline, perturbed, catch, and de-adaptation trials (Fig 1B). The experiment began with a baseline block of 15 unperturbed trials, followed by a perturbation block consisting of a series of perturbed trials during which the participants were required to perform 60 successful trials. This was followed by a single catch trial, where the perturbation was turned off without the participants' awareness, and then by an additional five perturbed trials. The experiment concluded with a de-adaptation block of 15 unperturbed trials, where the participants were made aware that the perturbation had been turned off. To show the computational plausibility of our hypothesis, we developed a computational model and simulated the same experimental protocol as in the human experiments. The computational model has feedback gain tuning, internal model learning and movement policy learning integrated in a hierarchical learning system under ecological control. The model was tested with a simplified dynamic model of the human body.

For each trial, we obtained a COM trajectory based on the kinematic data and labelled it as a success or a failure (Fig 1C). A trial was considered a success if the participant successfully performed the stand-up motion. Conversely, the trial was considered a failure if the participant made a corrective step to avoid falling over. While corrective steps are classified as failures within the constraints of our study, it should be noted that they can be a key strategy for maintaining stability in natural movements [28–30]. The overall effect of motor learning was assessed by calculating the trajectory area of the COM across the entire stand-up motion (Trajectory Area). The effect of motor planning was assessed by calculating

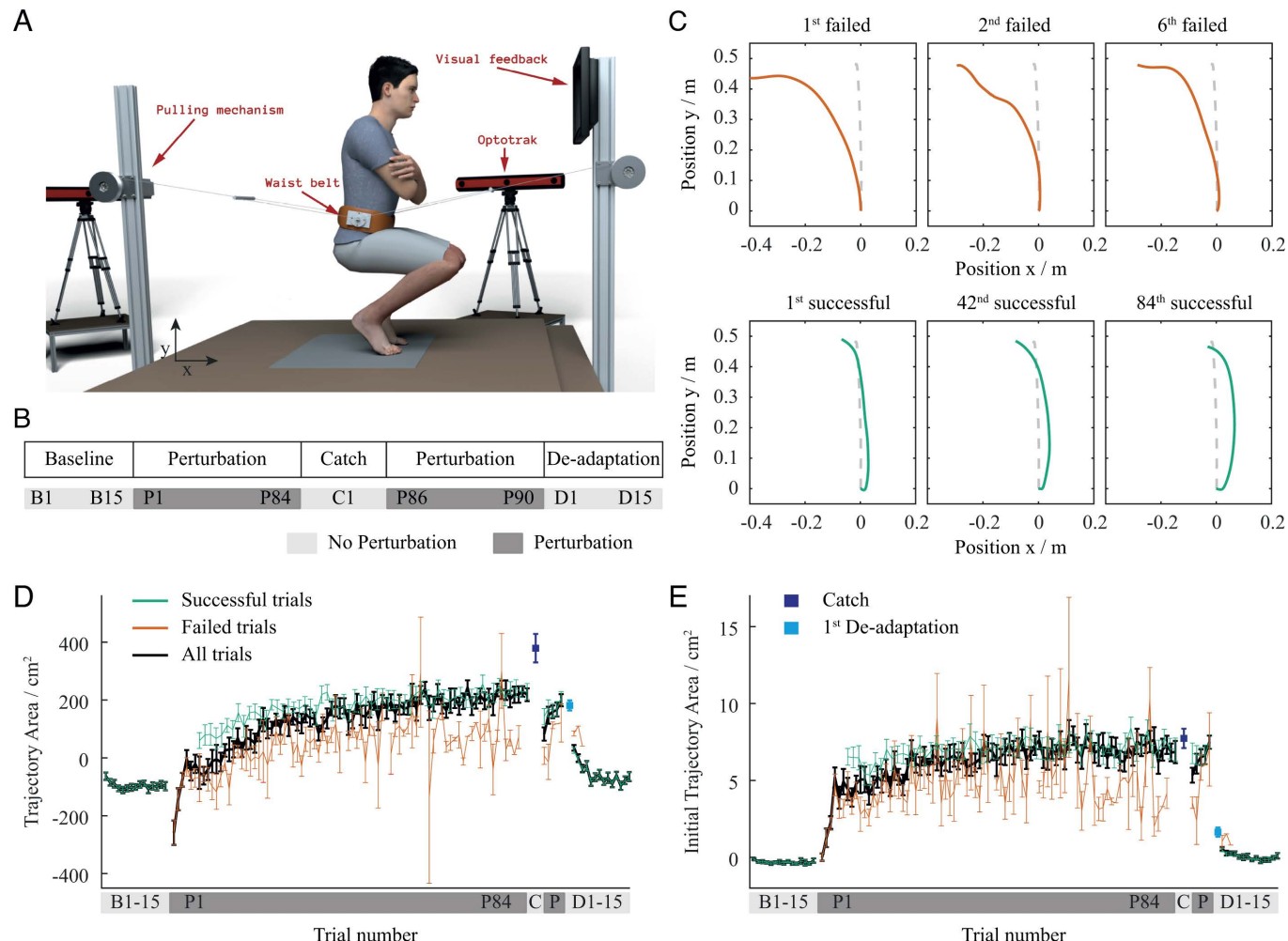

**Fig 1. Experimental setup and centre of mass adaptation dynamics.** (A) Illustration of the experimental setup where participants perform squat-to-stand movements using a custom force-controlled pulling mechanism to perturb their centre of mass. (B) Overview of the experimental protocol where light grey indicates unperturbed trials while dark grey indicates perturbed trials. (C) Failed and successful trajectories of a representative participant where dashed lines show baseline trajectory, red represents failed trials and grey represents successful trials. (D, E) Trial-by-trial progression of Trajectory Area and Initial Trajectory Area during the whole experiment where red represents failed trials, green represents successful trials, black represents all trials, dark blue represents the catch trial, and light blue represents the first de-adaptation trial. Error bars show standard error of the mean.

trajectory area in the initial phase of the motion, well before the feedback mechanisms could alter the motion (Initial Trajectory Area). The Trajectory Area and Initial Trajectory Area were defined as the integral of the deviation of the movement trajectory from a straight line in the sagittal plane. The reference line was drawn from the initial COM height in the squat position to the final COM height in the standing position for Trajectory Area. Initial Trajectory Area was calculated during the first 5 % of the motion, which occurred on average within (Mean±SD) 96.4±7.0 ms and corresponded to a height change of approximately 2.5 cm. Trajectory area calculation is illustrated in S1 Fig. The trial-by-trial progression of the average Trajectory Area and Initial Trajectory Area, for both successful and failed trials during the whole experiment, is shown in Fig 1D and 1E while the trial-by-trial progression of the COM forward and backward peak displacement is shown in S2 Fig. Moreover, to determine the relation between the feed-forward and feedback aspects of motor control, we calculated the Smoothness of the COM trajectory, and the Co-Contraction of the thigh antagonistic pair of muscles. Specifically,

we used Smoothness to assess the accuracy of the internal model [31–33] and Co-Contraction to assess the contribution of feedback on motor control [19,34]. It should be acknowledged that the level of muscle co-contraction has an effect on the jerk of the COM trajectory, and thus also influences Smoothness.

Participants began the experiment by performing 15 unperturbed stand-up motions. These trials revealed near-straight COM trajectories. Both the Trajectory Area and Initial Trajectory Area of these trials were consistently close to zero (B trials in Fig 1D and 1E).

In our experiment, failures were an integral part of motor adaptation. When perturbation was first introduced in the perturbation block, none of the participants were able to successfully perform the stand-up motion. The perturbation force pulled the participants backwards, which caused postural instability and required them to make at least one backward step to prevent falling over. Due to this backward motion, the movement trajectory of the early adaptation trials shifted in the direction of perturbation, resulting in a negative Trajectory Area (first few P trials in Fig 1D). On average, the participants made 6.15±3.10 failures before they could successfully perform the first stand-up motion.

To successfully perform the stand-up motions, and compensate for the perturbation force, the participants were required to adapt their usual stand-up motion by leaning against the anticipated perturbation. Even after their first successful trial, failures still sporadically occurred throughout the adaptation process. The participants made an average of 24.35±11.28 failures. The average number of failures significantly dropped in the early trials with an exponential learning rate of 15.7 trials and reached the plateau at trial P40.74±19.53. The adaptation process was reflected in the gradual increase of both Initial Trajectory Area and Trajectory Area measures in the direction against perturbation (trials P1 to P90 in Fig 1D and 1E). The Trajectory Area increase demonstrates the overall effects of motor learning, while the increase in Initial Trajectory Area shows the effect of motor planning. On average, the participants needed to perform 84.35±11.28 perturbed trials to reach the required 60 successful trials. At the end of the learning period participants successfully learned to adapt to the perturbation, indicated by the disappearance of failed trials and the prevalence of successful trials.

A single catch trial followed the 60 successful trials and none of the participants could successfully perform the stand-up motion in this condition. The Trajectory Area of the catch trial considerably increased in the direction against perturbation (dark blue dot of C trial in Fig 1D). After the perturbation block, the participants finished the experiment by performing 15 de-adaptation trials. Most participants failed to successfully perform the stand-up motion in the first de-adaptation trial, indicating that the participants were not able to instantly readapt to the unperturbed condition, despite the fact that they were consciously aware of it. Finally, both Trajectory Area and Initial Trajectory Area gradually returned to the same levels as in the baseline block (D trials in Fig 1D and 1E).

## Ecological control: success-efficient/failure-safe learning strategy

To interpret the role of failed motion execution in the process of overall adaptation, we investigated how stand-up motion was influenced by preceding failed trials compared to successful ones.

The Trajectory Area changes were significantly greater after failed trials than after successful trials ($t(19) = 2.92$, $p = .009$; Fig 2B). There was also a difference in changes in Initial Trajectory Area between trials following failure and trials following success, however this difference was not statistically significant ($t(19) = 1.62$, $p = .122$, Fig 2C). Specifically, after a failed trial, the COM trajectory bent in the direction against perturbation compared to the COM trajectory before a failed trial (Fig 2D). This resulted in an Initial Trajectory Area increase of the following successful trial, with respect to the successful trial performed prior to the failure ($t(19) = 2.23$, $p = .038$). Similarly, the Trajectory Area of the successful trial following a failed trial increased in respect to the successful trial prior to the failure ($t(19) = 3.20$, $p = .005$). This corresponds to a change towards the optimal COM trajectory where postural stability is maximised to prioritize safety (blue lines in S3 Fig). In this case, the optimal trajectory is bent in the direction against perturbation, so the Trajectory Area and Initial Trajectory Area are both larger compared to the optimal trajectory for maximal movement efficiency. On the other hand, the COM trajectory straightened during consecutive successful trials (Fig 2A). This resulted in Trajectory Area and Initial Trajectory

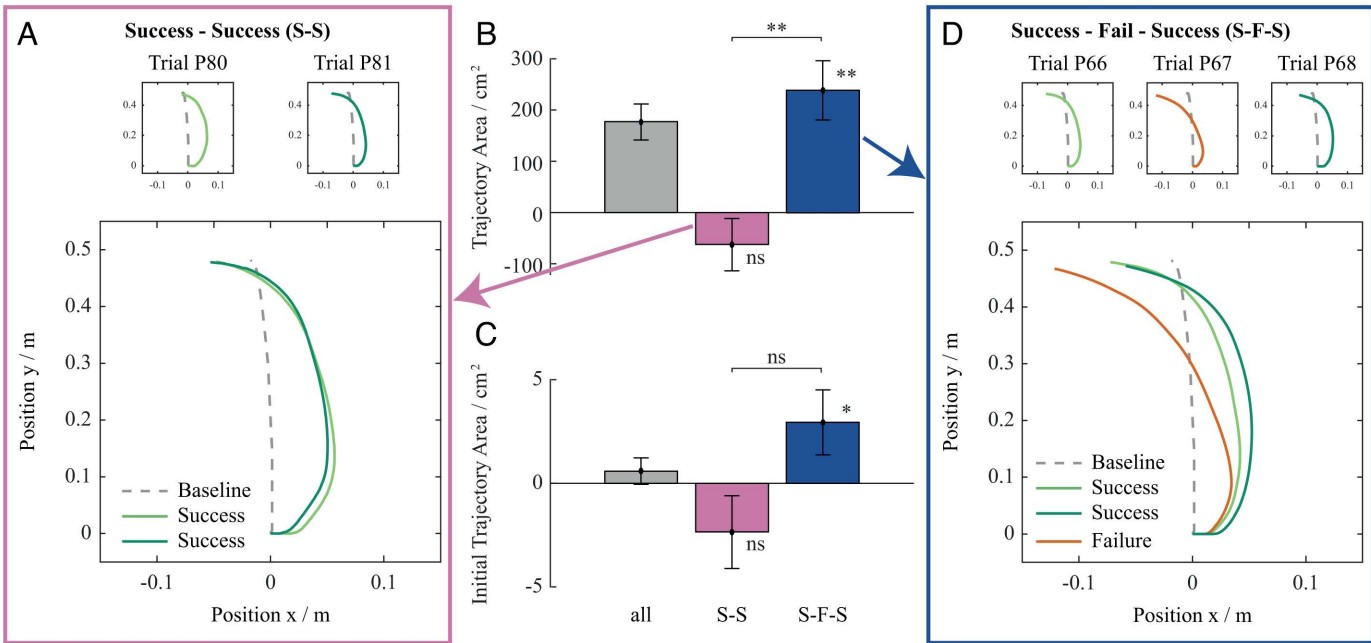

**Fig 2. Representative centre of mass trajectories and adaptation metrics.** (A) Representative set of centre of mass trajectories of two consecutive successful trials. (B, C) Total changes of Trajectory Area and Initial Trajectory Area, with changes between two consecutive successful trials (S-S) shown in pink and those between two consecutive successful trials separated by at least one failed trial (S-F-S) shown in blue. Error bars show standard error of the mean. (D) Representative set of centre of mass trajectories before and after a failed trial with successful trials shown in green and failed trials shown in red.

Area decreases during the consecutive successful trials, though neither change was statistically significant ($t$(19) = 1.22, $p$ = .239, $t$ = 1.57, $p$ = .134). The direction of this effect corresponds to a change towards optimal COM trajectory for maximal movement efficiency, where energy of motion is minimised (orange lines in S3 Fig). In this case trajectory is straightened and both the Trajectory Area and Initial Trajectory Area are reduced relative to the optimal trajectory for the maximal safety. Thus, overall, the participants adapted their movement patterns towards safe-optimal or energy-optimal, depending on whether they experienced a failure or a success.

To further emphasize the strong effect of failed trials beyond their overall contribution to adaptation, we analysed the average changes in Initial Trajectory Area and Trajectory Area across different trial sequences: between two consecutive successful trials, between two consecutive successful trials separated by at least one failure, and between consecutive failed trials. On average, during consecutive successful trials, the Initial Trajectory Area and Trajectory Area decreased by -0.05 ± 0.04 cm² and -1.51 ± 1.14 cm², respectively. In contrast, in a successful trial that followed a failure, the Initial Trajectory Area and Trajectory Area increased by 0.37 ± 0.18 cm² and 26.07 ± 8.59 cm², respectively, compared to the last successful trial before the failure. For consecutive failed trials, the average per-trial change was 0.53 ± 0.13 cm² for Initial Trajectory Area and 25.03 ± 10.00 cm² for Trajectory Area, reinforcing the role of failures in driving adaptation.

## Motor plan learning

We observed quicker learning of the motor plan compared to the effects of the overall motor learning. This is reflected in a different rate of adaptation between the Initial Trajectory Area and Trajectory Area ($t$(19) = 2.279, $p$ = .034). In particular, the Initial Trajectory Area significantly increased in early trials, with an exponential learning rate of 12.6 trials and a plateau at trial P34.35 ± 27.89 (black line in Fig 3C). The Trajectory Area increased over a longer period of time, with an

exponential learning rate of 20.3 trials, and a plateau at trial P50.61±23.90 (black line in Fig 3B). This shows that motor plan adaptation was completed while the overall adaptation to the perturbation was still ongoing. The motor plan no longer changed after the failures significantly decreased, while the adaptation of the whole COM trajectory continued even when failures became less frequent. Moreover, the motor plan only changed during failed trials and remained largely unchanged during successful trials. The Initial Trajectory Area increased during failed trials, with an exponential learning rate of 8.6 trials and a significant difference between the first and the last trials ($t(19)$ = -5.83, $p < .001$; red line in Fig 3C). It remained unchanged during successful trials, with no significant difference between the first and the last trials ($t(19)$ = -1.69, $p = .107$; green line in Fig 3C). On the contrary, overall adaptation was pronounced during both failed and successful trials. The Trajectory Area increased during failed and successful trials, as indicated by a significant difference between the first and the last trials (failed trials: $t(19)$ = 7.14, $p < .001$, red line in Fig 3B; successful trials: $t(19)$ = 5.37, $p < .001$, green line in Fig 3B). The exponential learning rates were 15.6 and 20.3 trials, respectively. Moreover, quicker effects of motor plan learning compared to the effects of the overall motor learning were also observed in the de-adaptation period. In particular, more trials were necessary for the Trajectory Area to return to baseline values than for the Initial Trajectory Area ($t(19)$

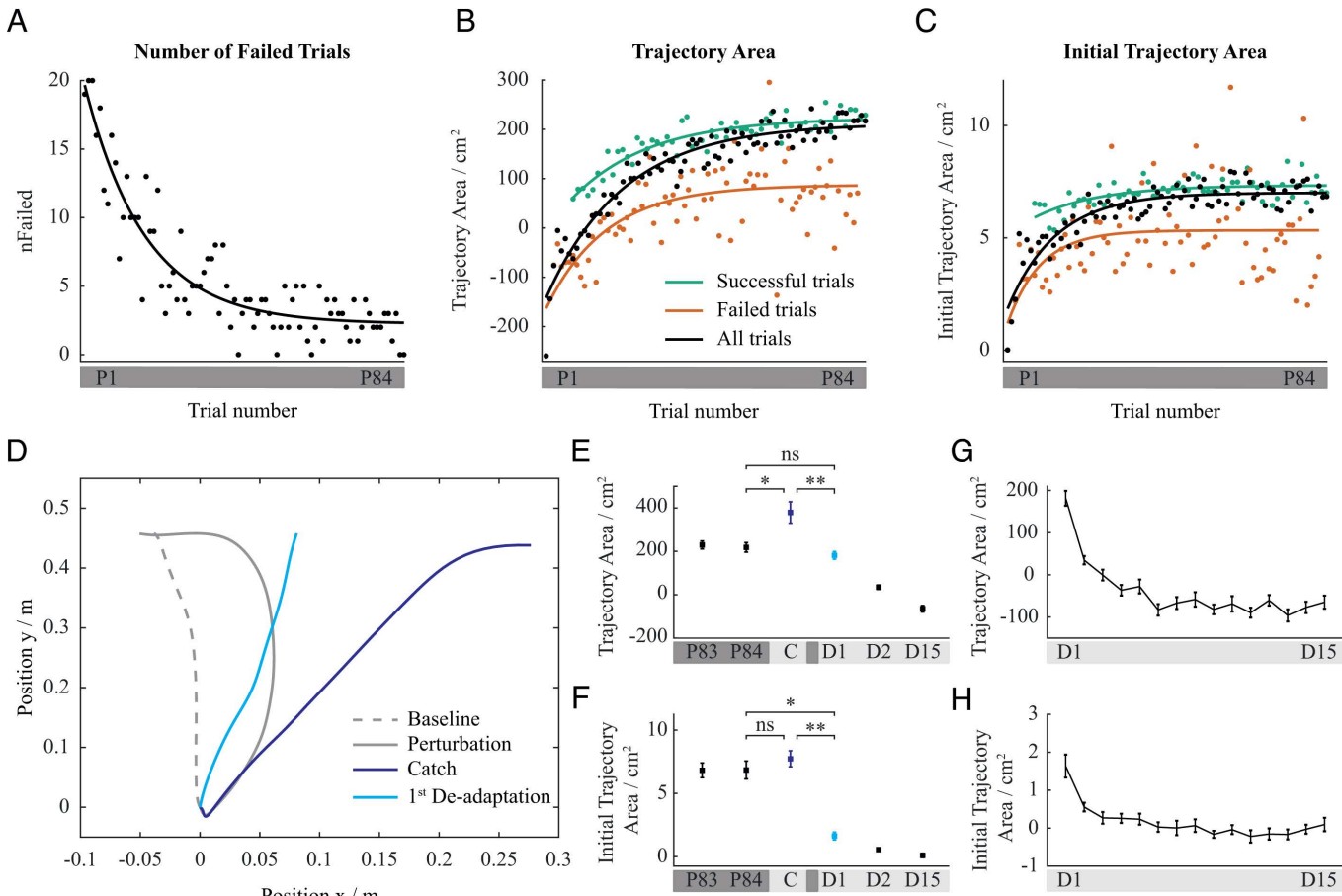

**Fig 3. Adaptation dynamics during perturbation and de-adaptation.** (A) The number of participants among the total of 20 that failed at each trial. (B, C) Progression of Trajectory Area and Initial Trajectory Area during adaptation to perturbation where red represents failed trials, green represents successful trials, and black represents all trials. (D) Representative set of centre of mass trajectories for unperturbed trial, for last perturbed trial, for catch trial and for first de-adaptation trial. (E, F) Comparison of Trajectory Area and Initial Trajectory Area of last perturbed trial, catch trial, and first de-adaptation trial. (G, H) Progression of Trajectory Area and Initial Trajectory Area during the de-adaptation block. Error bars show standard error of the mean.

= 3.970, $p < .001$; Fig 3G and 3H). Specifically, the Trajectory Area reached the baseline plateau after $12.48 \pm 3.79$ trials, while the Initial Trajectory Area required only $7.13 \pm 4.86$ trials.

To further elucidate the mechanism of high-level motor planning, we compared the motion during the catch trial with the motion during the first de-adaptation trial (Fig 3D). Specifically, this allowed for an assessment of the effect of participant awareness of upcoming perturbation-free trials. Participants made no change to their motor plan when they were unaware that the perturbation will be absent. However, when they were informed about the absence of the perturbation, their motor plan changed. There was a significant difference in the Initial Trajectory Area between the last perturbed trial, the catch trial and the first de-adaptation trial ($F(2, 38) = 10.57$, $p < .001$). Specific pairwise comparisons revealed that the Initial Trajectory Area of the catch trial was significantly larger than that of the first de-adaptation trial ($t(19) = 8.78$, $p < .001$). The Initial Trajectory Area of the catch trial was similar to the preceding perturbed trial ($t(19) = 0.37$, $p = .716$), while it significantly decreased during the first de-adaptation trial ($t(19) = 10.14$, $p < .001$), almost reaching the level of Initial Trajectory Area during unperturbed motion (Fig 3F). Consequently, there was also a difference on the level of the whole trajectory between the perturbed, catch and first de-adaptation trials ($F(2, 38) = 3.41$, $p = .044$). This is reflected in a significant difference of the Trajectory Area between the catch and first de-adaptation trial ($t(19) = 3.69$, $p = .0015$). Trajectory Area of the catch trial increased with respect to the preceding perturbed trial ($t(19) = 2.77$, $p = .012$), while it remained approximately the same during the first de-adaptation trial ($t(19) = 1.89$, $p = .074$; Fig 3E). Moreover, the relative reduction in the first de-adaptation trial with respect to the catch trial was significantly larger for the Initial Trajectory Area ($0.69 \pm 0.34$) than for Trajectory Area ($0.33 \pm 0.29$) ($t = 4.585$, $p < .001$). This demonstrates quicker changes on the level of the motor plan compared to the changes on the level of the whole trajectory.

## Internal model learning

To understand the adaptation process distinctive from the adaptation of the plan, we further investigated the adaptation of the lower-level feed-forward motor control. Internal model changes during the adaptation to perturbation are reflected in aftereffects during catch and de-adaptation trials. None of the participants could successfully perform the stand-up motion during the catch trial, and most participants failed to successfully perform the stand-up motion in the first de-adaptation trial. The Trajectory Area of the catch trial was considerably larger than the Trajectory Area of the perturbed trials (Fig 3E). Moreover, the Trajectory Area of the first de-adaptation trial was larger compared to the unperturbed trials, despite the Initial Trajectory Area of this trial being noticeably smaller than the Initial Trajectory Area of the preceding perturbed trials, and similar to the Initial Trajectory Area of unperturbed trials (Fig 3F). Nevertheless, despite the motor plan during the de-adaptation trial being changed appropriately with respect to no anticipated perturbation, participants still mostly failed to successfully perform the stand-up motion.

Improvement of the internal model during the adaptation to perturbation is further reflected in a progressive increase of Smoothness over the whole perturbation block with an exponential learning rate of 29.4 trials (black line in Fig 4A), and a plateau at trial $P58.13 \pm 26.11$. This is significantly slower than the reduction of the number of failed trials ($t(15) = 4.61$, $p < .001$) and the related adaptation of the motor plan ($t(15) = 2.96$, $p = .010$). Moreover, Smoothness and its rate of change is comparable for both successful and failed trials (Fig 4A), indicated by no statistically significant difference in Smoothness between the two ($t(19) = 1.01$, $p = .324$).

## Feedback gains are manipulated based on success and failure

In line with internal model improvements, a concomitant decrease in feedback contribution was observed. Co-Contraction plateaued at trial $P67.42 \pm 17.70$, which was not significantly different to the plateau of Smoothness ($t(11) = 1.43$, $p = .179$). The trials at which all adaptation measures reached plateaus are shown in S4 Fig, and their statistical comparison is on S1 Table. Although the contribution of feedback steadily decreased during the adaptation to perturbation, this was notable only during consecutive successful trials. Feedback contribution during failed trials remained high. Co-Contraction of failed

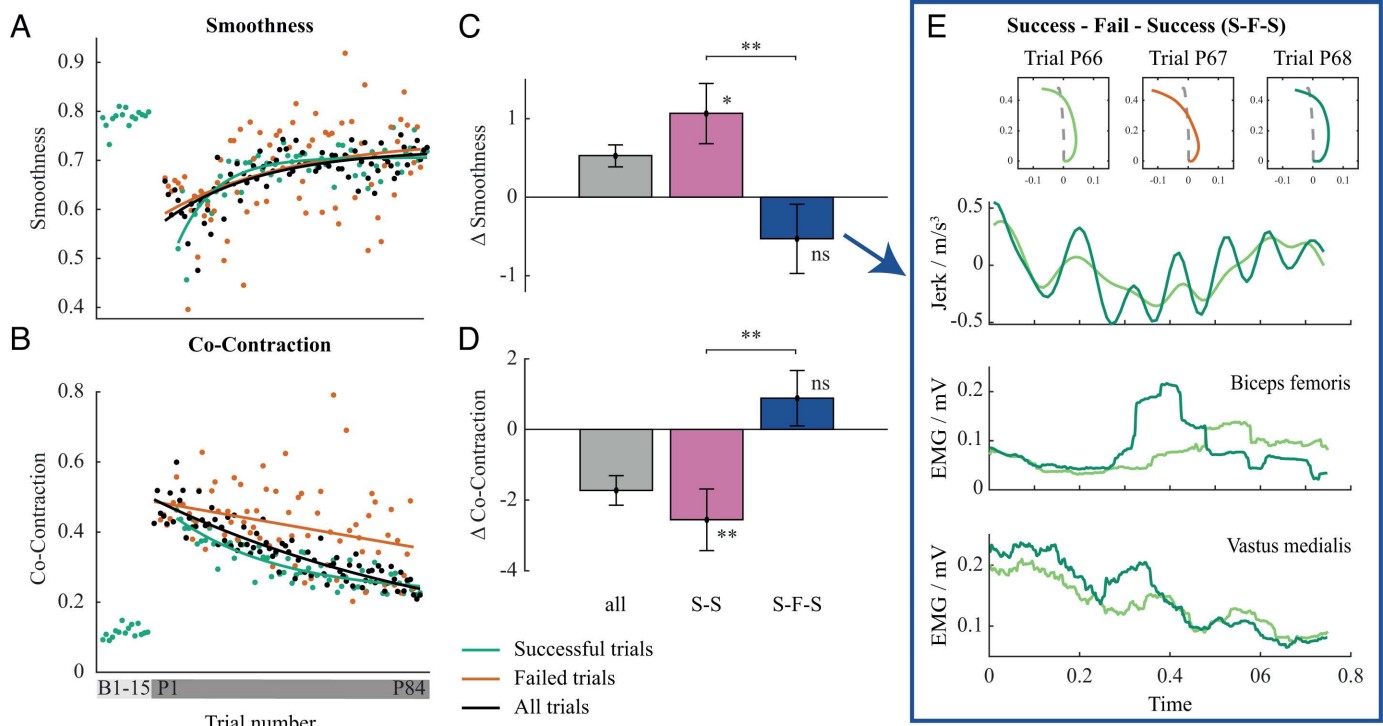

**Fig 4. Progression of Smoothness and Co-Contraction during adaptation to perturbation.** (A, B) The progression of Smoothness and Co-contraction during adaptation to perturbation, with red representing failed trials, green representing successful trials, and black representing all trials. (C, D) Total changes of Smoothness and Co-Contraction where change between two consecutive successful trials (S-S) is shown in pink and change between two consecutive successful trials separated by at least one failed trial (S-F-S) is shown in blue. (E) Representative jerk of centre of mass trajectory and EMG trajectories for successful trials before a failed trial (light green) and after a failed trial (dark green). Error bars show standard error of the mean.

trials throughout the adaptation to perturbation was high (red line in lower panel of Fig 4B), there was a significant difference between the mean Co-Contraction of successful and failed trials ($t(19) = 3.22$, $p = .005$), and no significant difference of Co-Contraction between the first failed and last failed trials ($t(19) = 1.32$, $p = .204$).

Contribution of feedback decreased only during consecutive successful trials and it increased when a failure occurred. This is reflected in the statistically significant decrease in Co-Contraction during consecutive successful trials ($t(19) = 3.18$, $p = .005$). On the other hand, the Co-Contraction of a successful trial after failure increased compared to a successful trial before a failure (Fig 4D). However, the change was not statistically significant ($t(19) = 1.23$, $p = .235$). Increase of feedback gains after failed trials, reflected by Co-Contraction increases, also caused the Smoothness to decrease. However, the change was not statistically significant ($t(19) = 1.20$, $p = .245$). On the other hand, there was a statistically significant increase of the Smoothness during the consecutive successful trials ($t(19) = 2.78$, $p = .012$; Fig 4C).

## Computational justification

The goal of the computational modelling is to show that the "Success-efficient/failure-safe Ecological Control", with hierarchical reinforcement learning and internal model learning form a mechanistically valid set of procedures that reproduce the human experimental results. In principle, the adaptation patterns and the similarity of the model-generated data to that of the human participants', would corroborate the conclusions derived from the experimental data. With this logic, we implemented a modular and hierarchical learning architecture under Ecological Control, (Fig 5) using Matlab (2019, The

**Fig 5. Proposed Computational Model of Hierarchical Motor Learning.** The model integrates reinforcement learning, internal model adaptation, and feedback control to simulate human motor learning under ecological control. Reinforcement learning optimizes motor plans based on trial outcomes, the internal model adapts to new body-environment dynamics, and feedback control compensates for inaccuracies in inverse dynamics. The model follows the experimental protocol, enabling direct comparison with human data and capturing key aspects of adaptation, including trial-by-trial learning and the role of failures in shaping movement strategies.

MathWorks Inc., Natick, USA) in a computer environment. The computational model was implemented such that it can systematically execute the experimental protocol that was defined for the human experiment. As such, each simulated experiment of the computational model could be considered as a simulation of the learning attempt of a human partici-pant, with each execution generating varying data as the model is stochastic. This allows synthetic but realistic population statistics to be computed. Accordingly, a series of simulated experiments ($N=60$) with the computational model was con-ducted, and compatibility of the simulated results with the experimental results was assessed. In the following paragraphs, "model" refers to simulated results while "human" refers to the experimental results.

The model succeeded to complete the full experimental paradigm in 59 out of 60 cases. In one case the simulated body could not learn to stand up in the allotted maximum number of steps, which is not included in the data analysis concerning the model simulation. Similar to humans, the early failures of the computational model were backward falls in the direction of perturbation causing negative Trajectory Area values (compare simulated results in Fig 6A with the early P trials of the experimental results in Fig 1D). To stand up without a fall, the model adapted the motor plan by reinforcement

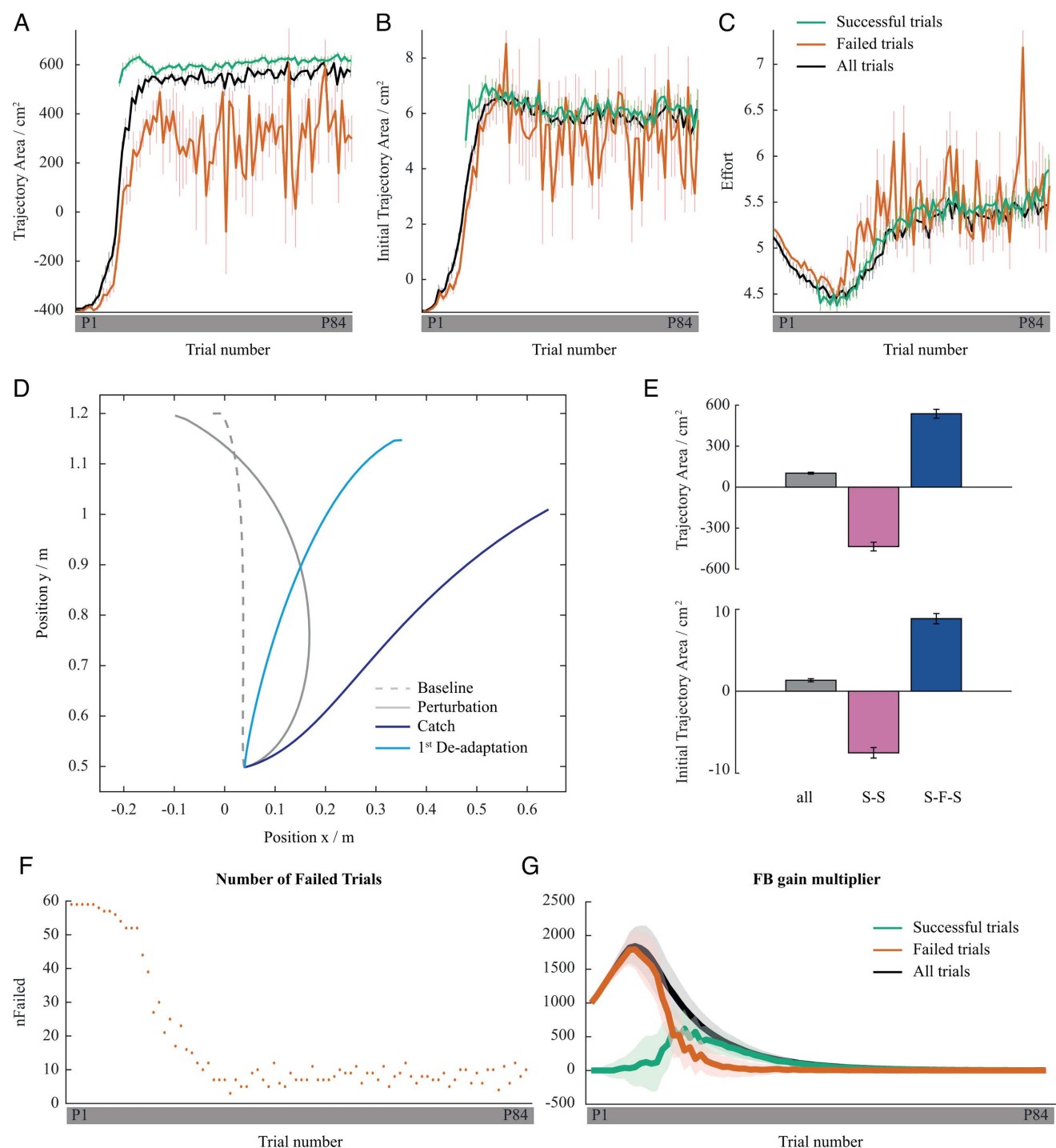

**Fig 6. Simulation results of ecological and hierarchical motor learning model.** (A, B, C) Trajectory Area, Initial Trajectory Area, and Effort along normalised learning trials where red represents failed trials, green represents successful trials, and black represents all trials. (D) Representative set of simulated centre of mass trajectories for unperturbed trial, for last perturbed trial, for catch trial and for first de-adaptation trial. (E) Trajectory Area and Initial Trajectory Area changes over simulated experiments for 100 simulated participants for all trials, for two consecutive successful trials (S-S), and for consecutive successful trials separated by at least one failure (S-F-S). (F) Average number of failures among the simulated N = 59 participants at each trial number. (G) Average feedback gain multiplier as a function of perturbed trial number for failed, successful, and all trials.

learning as well as learning the novel dynamics created by the perturbation with an internal model by using prediction error feedback. The effect of learning in the model was reflected by a COM trajectory shift in the direction against perturbation, and gradual increases in Initial Trajectory Area and Trajectory Area measures which is in line with the human experiments (compare simulated results in Fig 6A and 6B with the experimental results in Fig 1D and 1E).

In the model, the failures caused movement policy updates to reduce the probability of falls. In parallel, the internal dynamics model learned the new dynamics so that the policy could be accurately executed. This resulted in data similar to that of humans: the model made an average of $26.90 \pm 12.96$ failures to complete the experiment (humans: $24.35 \pm 11.28$). In the simulated experiments, even after the first successful trial (reached in $15.73 \pm 2.55$ trials on average), failures still occurred throughout the adaptation process as in human participants. Moreover, the number of failures sharply reduced after the first 20 trials similar to the learning pattern observed in the human experiments. In the case of the model, the failures after the first success were due to the search for a movement-efficient solution once a 'confidence'had been developed for safely standing up. This caused a slight increase in the number of falls after this 'confidence'was attained.

To quantitatively assess how well the model captures human behavior, we computed the Pearson correlation coefficient between the simulated model-generated Trajectory Area and Initial Trajectory Area data and the corresponding data from human participants. The model performed 59 simulation runs, while participants collectively generated 20 experimental runs. Due to the stochastic nature of the model, each run produces different behaviors. The goal was to determine how frequently the simulator generates behaviors comparable to those of human participants. This resulted in a total of 1,180 ($59 \times 20$) pairs of Trajectory Area and Initial Trajectory Area data subjected to correlation analysis.

The results indicate that the model captures the Trajectory Area patterns of participants to a high degree. Among the 1,180 participant-simulation run pairs, only 26 (2.2 %) failed to show a significant positive correlation ($p < .05$), with 16 of these cases coming from participant 7, whose pattern is the least well captured by the model. The upper pane of S5 Fig illustrates the correlation coefficients across simulation runs for each participant, showing minimum, maximum, and mean values. The number of simulation runs in which a participant did not exhibit a significant correlation is also indicated in parentheses. It can be observed that almost all participants' Trajectory Area data significantly correlate with the model-generated data, with correlation values ranging from.403 (participant 1) to.841 (participant 9). Across participants, the highest correlation observed for each individual results in a mean correlation coefficient of.63.

When analyzing the Initial Trajectory Area, the number of participant-simulation run pairs failing to show a significant correlation increased to 340 out of 1,180 (28.8 %), indicating a weaker correspondence compared to the full Trajectory Area analysis. The lower pane of S5 Fig illustrates the correlation coefficients across simulation runs for each participant, again showing minimum, maximum, and mean values. While the model captures Trajectory Area patterns well, it also captures Initial Trajectory Area patterns to a considerable extent. Except for participant 6, all participants' Initial Trajectory Area data show significant correlations with at least some simulation runs, with correlation values ranging from.215 (participant 13) to.73 (participant 9). The results in the lower pane of S5 Fig show that for 15 out of 20 participants, their Initial Trajectory Area data significantly correlate with at least two-thirds of the simulation data. Across participants, the highest correlation observed for each individual results in a mean correlation coefficient of.52, excluding participant 6, who had no significantly correlated simulation runs.

Overall, the correlation analysis suggests that the model effectively captures the general Trajectory Area behavior of human participants, with the vast majority of participant-simulation pairs showing significant correlations. While the model also captures Initial Trajectory Area patterns, these correlations are somewhat weaker, likely due to greater variability in initial movement dynamics among participants.

## Success-efficient/failure-safe learning strategy guiding reinforcement learning

In the computational model, reinforcement learning searches for motor plans so as to maximise total reward, whereas the lower-level motor adaptation learns an internal model of the body-environment. On the very top, Ecological Control determines which reward function must be used by reinforcement learning. According to our hypothesis,

failure triggers risk aversion and switches learning to 'safe mode'. When this logic is implemented in the model, it is able to replicate participant behaviours. When humans failed a trial, on average, Initial Trajectory Area of the following successful trial increased relative to the successful trial prior to the failure. Conversely, Initial Trajectory Area decreased during the consecutive successful trials (Fig 2C). This pattern is well replicated by the computational model. In the simulated experiments, Initial Trajectory Area and Trajectory Area decreased during consecutive successful trials, whereas both parameters increased in the success trials following a failure, relative to the preceding successful trial (Fig 6E). This suggests that the Ecological Control mechanism within the computational model that guides reinforcement learning (i.e., the arbitration of two rewards functions, one favouring safety and the other favouring effort minimization), based on immediate task performance and task proficiency, captures the mechanisms of the human central nervous system in dealing with full body perturbations where bodily injury must be constantly monitored.

## Interplay between internal model learning and motor plan learning

The human experiments demonstrated that Initial Trajectory Area significantly increased only during failed trials and remained largely unchanged for successful trials. The same pattern was found with the computational model, indicated by Initial Trajectory Area remaining consistent throughout the successful trials, and differing significantly during the failed trials. Likewise, the computational model produced compatible simulation results with respect to the human data showing that Trajectory Area increased during both failed and successful trials with a significant difference between the first and the last trials.

In line with the experimental protocol, the computational model was subjected to catch and de-adaptation trials. To emulate the catch trial in simulation, the perturbation force was simply set to zero. For the de-adaptation condition, the conscious knowledge of the participants on the upcoming unperturbed trials was modelled by: (1) resetting the high-level plan to a 'memory' of a plan that had yielded a successful unperturbed trial execution earlier, and (3) shifting the internal model parameters towards the no-perturbation levels (see S1 and S2 Algorithm files).

In the catch trial, the body under computational model control made a forward fall in the direction against perturbation with a largely curved COM trajectory (blue curve in Fig 6D). This produced a large Trajectory Area, as observed in the human participants. The large Trajectory Area was due to learned internal model generating ankle torques in the direction against perturbation to counteract the expected perturbation force, which has been absent during the catch trial. The Initial Trajectory Area changed considerably less compared to the Trajectory Area paralleling the COM trajectory pattern observed in humans in the catch trials.

The performance of the model during the de-adaptation trials was also compatible with that of human participants, as Trajectory Area and Initial Trajectory Area values gradually returned to the same levels as in the baseline block. The simulated body controlled by the model failed to stand up in the first de-adaptation trial, which was the case for most of the human participants. The Initial Trajectory Area in the first trial was smaller than the previous perturbed movement, remaining between the Initial Trajectory Area values of unperturbed and perturbed trials (compare light blue and grey curves in Fig 6D). Trajectory Area, however was much larger than the unperturbed trials (compare light blue and dashed grey curves in Fig 6D). These two observations are in agreement with the human data. In particular, the Initial Trajectory Area of the first de-adaptation trial was smaller than the Initial Trajectory Area of the preceding perturbed trials and similar to the Initial Trajectory Area of the unperturbed trials (light blue dot of trial D1 in Fig 1E). Furthermore, the Trajectory Area was larger compared to unperturbed trials (light blue dot of trial D1 in Fig 1D).

The catch and de-adaptation trial results indicate that perturbation compensation with feedback control alone is not sufficient for task completion. Likewise, a mere voluntary change in plan alone is also not sufficient to complete the task as the internal model cannot be completely switched back to the unperturbed state instantaneously.

### Feedback gains are manipulated based on success and failure

While human participants were learning how to stand up in the perturbation condition, their Co-Contraction steadily decreased. This may be attributed to the improvement of the internal model, and the related decrease of the feedback contribution to the motor control (Fig 4B). However, this is notable only when participants had consecutive successful trials, as Co-Contraction increased when a failure occurred (Fig 4D). Although in the computational model Co-Contraction was not considered, the feedback gains showed a specificity with regard to the successes and failures which generated a feedback gain pattern similar to the Co-Contraction patterns of the human participants. In particular, in the simulated experiments, after an initial increase during the first 10 trials on the average, the feedback gain multiplier decreased and became almost zero after the 40th trial on the average (black curve in Fig 6G). Similar to the Co-Contraction increase in humans during failures, the simulated experiment data showed that the model failures well coincided with high feedback gains. Note that during successful trials the feedback multiplier was low even before the 40th trial (red and green curves in Fig 6G). Importantly, the feedback contribution was implemented as a scalar gain multiplier acting on fixed proportional (P) and differential (D) gains, which was regulated in relation to internal model prediction error regardless of whether the model experiences a failure or a success.

Overall, our computational model is able to reproduce a large portion of human data indicating that the proposed hierarchical modular learning system under Ecological Control is a plausible learning architecture that can be adopted by the human central nervous system.

## Discussion

We investigated whole-body motor adaptations where a risk of failure was an integral part of the learning process. In contrast to arm-reaching studies where failed trials are classified as trials when the participant did not successfully reach the desired target [35], failures in our experiment are movements when the participant lost postural balance and was required to make corrective steps to prevent a fall. In our study, a considerable number of failed trials were observed during learning. At the end of the learning period, participants successfully learned to adapt to the perturbation resulting in the elimination of failed trials and the prevalence of successful trials. This is in line with previous research where the number of failed trials decreased during arm-reaching [35] and whole-body motor learning [36].

### Balancing safety and efficiency through trial-based reinforcement learning

Motor behaviour was predominantly adapted after failed trials which clearly shows reinforcement learning mechanisms [37–40]. After the failed trials, the COM trajectory shifted in the direction against perturbation resulting in a shift towards a safer and more stable movement pattern. Conversely, consecutive successful trials led to a straightened trajectory and resulted in a more efficient movement pattern. The behavioural results together with the computational study strongly support that trial-based task outcome is responsible for higher level plan adaptation to maximise ecological fitness through reinforcement learning [41–44]. Under this theory, failures push the plan towards safer execution of motion, while successful motions promote optimization of movement efficiency.

When changes in the environment are perceived as a potential danger during the execution of movements, risk-averse learning with prioritised safety takes place [45]. This allows for the learning process to avoid the risk of injury and in the long term leads to higher chances of survival [1]. Once the environment is no longer perceived as dangerous, cost functions that relate to the execution of the motion, such as minimization of effort, can be prioritized [22]. Only cautious execution of motion in dangerous situations (failure-safe) together with the minimization of effort in safe situations (success-efficient) can lead to truly optimised ecological fitness.

This success-efficient/failure-safe learning mode switching mechanism is similar to the win-stay/lose-shift strategy observed in neuroeconomics and social linguistics [46,47]. The win-stay/lose-shift strategy is a heuristic learning strategy to retain a successful action but to switch to another action after a failure [48]. In our experiment participants retained

and optimised their movements during successful motions, but switched to a safer movement after a failure. By switching between learning modes based on success and failure, individuals were able to quickly adapt to new motor tasks while minimising the risk of injury.

From the perspective of the exploitation-exploration trade-off in human decision-making [49] and reinforcement learning [37], our model suggests that successful motions encourage the exploration of more energy-efficient movements, whereas failures lead to a switch to a safe policy that ensures task completion without failure, thereby functioning as exploitation. In humans, the exploration-exploitation trade-off is dynamically adjusted and task-specific [50,51]. This trade-off can be modulated by context [52] and emotional state [53].

In tasks where there is no risk of injury, exploration is typically favored at the outset, while exploitation becomes the dominant strategy over time [50,54]. However, in response to unexpected failure or loss, exploration levels tend to increase [50,55,56], which contrasts with our task. We argue that this is due to the risk of injury and the high energy demands of our task, distinguishing it from other contexts where failure does not pose a risk of harm.

Several studies have shown that tasks learnable through error feedback can also be acquired via reinforcement feedback alone [57–60]. From a computational standpoint, error feedback provides richer information than reinforcement feedback and should be leveraged whenever available. This observation is supported by experimental findings (e.g., [57], although mixed results exist [58]. These studies also confirm that the central nervous system employs multiple learning strategies to solve sensorimotor learning tasks, and that they are recruited in a task-dependent way. It has been also showed that combining reinforcement and error-based learning improves skill acquisition and retention in the full-body movement of basketball shooting [60]. However, the interplay between these learning mechanisms and the task-dependent recruitment of reinforcement learning processes in the brain remains poorly understood [59,61]. We argue that these two learning processes operate hierarchically under ecological constraints, yet run independently and in parallel, enabling rapid responses essential for ecological fitness and survival.

In the complex task of controlling the body for safe movement under altered dynamics, the central nervous system must simultaneously learn to represent new body-environment dynamics while avoiding failure. The former is supported by prediction error feedback, which is continuously processed through sensorimotor loops in the central nervous system. The latter, however, relies on delayed feedback, only available at the end of a movement in the form of success or failure, necessitating reinforcement learning. Experimental evidence suggests that feedback error learning is mediated by the cerebellum, while reinforcement learning involves the basal ganglia and multiple brain structures encoding value functions [61,62], indicating a division of labor and modularity in the brain.

## Modularity and hierarchy in motor learning

We observed faster adaptation in motor plan learning compared with overall motor learning. While motor plan learning took place in early trials and was no longer changed after the failures significantly decreased, overall motor learning continued over a longer period of time even after the failures were no longer frequent (as in [35]). Similarly, in the de-adaptation period, the motor plan was more quickly changed, while it took longer for the overall trajectory to return to baseline values. This shows two distinctive plan and motor control processes with different rates of adaptation, faster on the level of planning and slower on the level of internal model learning, which indicates motor control mechanism modularity. This is in line with observations that motor adaptation processes in the brain rely on distinct neural systems that operate on different times-cales and retain information at different rates [63–66]. In particular, slow mechanisms retain motor memories on longer times scales, whereas fast mechanisms enable us to quickly adapt to potential transient changes in the environment [64,67]. Moreover, in our study, the motor plan significantly changed only during failed trials. This suggests that only failed execution of motion activates the adaptation mechanism to adapt the plan in order to maximise ecological fitness. On the contrary, the whole trajectory changed both during failed and successful trials. This further demonstrates that internal model adaptation is distinctive from the adaptation of the plan and is governed by optimality criteria not directly related to ecological fitness

(e.g., minimization of motor prediction error). Previous work suggests that the fast and slow processes of learning may be underpinned by explicit strategy learning and implicit motor adaptation [10,68]. Moreover, Ikegami et al. [35] have proposed that failure driven adaptation is an explicit strategy learning since it is faster than implicit components represented by internal model adaptation and is only active in the presence of failures [68,69]. Explicit strategy learning enables us to quickly adjust the motor plan to a suitable solution which ensures the safe execution of motion, while the optimization process persists [70].

To further elucidate the mechanisms and the relation between the two distinctive adaptation processes, we compared the motion during the catch trial with the first de-adaptation trial. This allowed us to assess the influence of anticipation, as participants were first unaware that the perturbation would be turned off (catch), and later informed that there would then be no perturbation (de-adaptation). Based on the previous arm-reaching studies [71], we would expect that despite the verbal instructions, participants would exhibit large aftereffects during the de-adaptation trial comparable in magnitude with the aftereffects of the catch trials. However, as our results indicate, the motor plan was significantly different between the two trials. While the motor plan for the catch trial remained similar to the preceding perturbed trials, it significantly changed during the first de-adaptation trial when the participants consciously made quick explicit changes to the motor plan as they were aware that the perturbation would be absent. A possible reason for the discrepancy between our results, and the results of previous arm-reaching studies, could lie in the ecological aspects of our experiment. Specifically, the cost of failure during the arm-reaching movement is negligible, but very pronounced in our whole-body movement where a failure would carry a substantial injury risk. Nevertheless, although the motor plan during the de-adaptation trial was changed appropriately with respect to absence of perturbation, participants still mostly failed to successfully perform the stand-up motion. This suggests that participants were not able to instantaneously re-adapt the internal model as they did for the motion plan. In this regard, our results are in line with previous work suggesting that implicit learning occurs without conscious awareness and can result in large and prolonged aftereffects which reflect changes to the internal model during previous adaptation [72]. This implies hierarchical organisation of the two adaptation processes. The motor plan was adapted on a higher level by conscious understanding of the motor task while on the lower level, the internal model could not be consciously changed and needed more time to re-adapt.

### Interplay of internal model adaptation and feedback contributions

Prominent aftereffects during catch and de-adaptation trials reflect adaptation on the level of internal model of movement dynamics. The aftereffects of force adaptation have been interpreted as evidence that the nervous system learns to anticipate and counteract novel forces by building an internal model of the body and the environment [11,15,16,71,73,74]. Improvement of the internal model during the adaptation to perturbation is further evident in a progressive increase of Smoothness over the whole perturbation block [75]. This suggests that, alongside the refinement of the motor plan, the internal model becomes more accurate in predicting and guiding smoother movements. The observed increase in Smoothness was slower than the rate of error reduction, indicating a more incremental refinement of the internal model [76]. Notably, the improvement of the internal model, which relies on time-sensitive sensory feedback, exhibits similar patterns in both successful and failed trials. This suggests that the internal model undergoes adaptive changes during both successful and failed attempts at motion execution. These results suggest that the adaptation of the internal model may occur at lower levels of motor control and could be independent of reinforcement learning mechanisms.

In contrast with the improvement of the internal model during both successful and failed motion, feedback contribution of the motor control decreased only during the successful motion execution. On the other hand, motor control feedback contribution remained high and largely unchanged during the failed trials, despite the improvement of the internal model. The effect of a failed trial was notable even in the subsequent successful motion, where the feedback contribution increased with respect to the successful motion before the failed trial. This could happen since, due to the recent failure, the internal model was not trusted so feedback contribution was amplified to increase robustness to perturbation and to successfully perform the motion [19,77,78].

## Computational perspective on success-efficient/failure-safe strategy

We developed a computational model that captures the trade-off between safety and movement efficiency through reinforcement learning with a modular and hierarchical organisation, and also includes internal model adaptation with feedback gain tuning. The model was simulated with a body model consisting of two degrees of freedom, to show that it is not a mere conceptual model but can indeed generate learning patterns similar to those observed in the experiments. In particular, the model could successfully reproduce (1) the trial outcome-dependent changes in motor plan and motor execution, (2) the contrast in the trajectories observed in catch versus de-adaptation trials, and (3) the reduction in the number of failures. One explanation as to why such a safety-efficiency switching hierarchical architecture is adopted by humans might lie in the concept of bounded rationality in sensorimotor learning [79–81]. This suggests that the brain, constrained by its imperfect internal models and limited computational capacity, cannot achieve optimal solutions through a singular mode of operation. For example, from a theoretical perspective, optimal feedback control can be used to solve any motor task without requiring a hierarchical organization and intermediate trajectory representation [21,22]; but from an ecological point of view, this would be prohibitively costly for an organism having a tight energy and time budget to survive. Therefore, without denying the possible role of optimal feedback control for tasks involving well learned limb dynamics such as in arm movements (e.g., [82]), we adopt a hierarchical view of motor control that makes learning computationally less intensive. In addition, unlike artificial learning systems, the wellbeing of the organism must be maintained at all times. Therefore, the optimality of the final solution cannot be the ultimate goal, as the cost incurred during learning must also be considered. Thus, the brain navigates the complex interplay of environmental uncertainties, internal noise, and the uncertain range of acceptable failures. Alternation between a conservative safe mode in the face of failure and a riskier movement efficient mode following success could be seen as an adaptive mechanism aimed at balancing safety with movement efficiency. This strategy allows for continual refinement of internal models, leveraging a degree of risk for potential movement efficiency gains while avoiding the dangers of excessive risk-taking. This adaptive switching can be viewed as the pragmatic approach of the brain to find satisfactory solutions amidst inherent uncertainties and computational limitations. When the model switches to effective mode after a sufficient number of successful trials, it explores the motor plan space towards finding energy efficient plans. This is possible because the reinforcement learning system dynamically constructs a motor plan distribution (represented as a Gaussian distribution with a mean and standard deviation) from which plans are sampled and executed. A more energy efficient execution with no failure triggers an update in the motor policy network so that the policy mean is shifted towards the newly discovered plan. This is akin to reward-dependent modulation of motor variability for implicit exploration that is often attributed to the basal ganglia [55]. Our model does not demonstrate an increased exploration when a failure occurs; instead, a safe solution is instantiated, and exploration ensues from that safe motor plan. Although it is reported in the literature that humans increase movement variability during low success or minimal feedback [55,83], we believe that this is task dependent: when bodily injury is at stake after a failure, first safety must be maintained and then exploration should be reinitiated. This is how our participants behaved in our experimental setup which guided our modeling.

## Limitations

The relationship between co-contraction, impedance control, and feedback control is critical for understanding motor adaptation mechanisms. Co-contraction is primarily associated with impedance control, which modulates joint stiffness and stability to handle perturbations (e.g., [84,85]). Feedback control, on the other hand, involves real-time adjustments to motor output based on sensory input errors, often modeled as part of an optimal control framework [21]. While co-contraction can indirectly influence feedback mechanisms through automatic gain scaling [86], it is not a direct proxy for feedback control. In this study, co-contraction was used as an indirect measure to infer changes in feedback gains rather than feedback control itself, leveraging prior evidence that feedback gains are scaled by muscle activity [86] and that co-contraction enhances feedback responses [87–90]. However, this interpretation omits the specific role of impedance control mechanisms that arise from

co-contraction. As demonstrated by Calalo et al. [91], the sensorimotor system modulates co-contraction relative to visuomotor feedback responses to regulate movement variability. Their findings underscore that co-contraction primarily functions to increase joint stiffness and reduce movement variability, whereas feedback control dynamically corrects for positional deviations. However, unlike our study, they measured visuomotor feedback responses directly, which allowed them to explore the relationship between co-contraction and feedback responses with greater precision. Our study lacks this direct assessment, limiting our ability to comprehensively characterize the interplay between these two mechanisms.

Moreover, our computational model, which implements proportional-derivative (PD) control, does not incorporate impedance control through co-contraction dynamics. This represents a notable limitation in aligning the model with the observed human behavior. Future work should aim to extend the model to include muscle-specific impedance control, potentially through simulations incorporating appropriate agonist-antagonist co-contractions. As highlighted by Franklin et al. [92], impedance control can be particularly advantageous when anticipatory strategies are required, allowing the system to maintain stability without relying solely on rapid sensory feedback. By integrating muscle-specific impedance control into computational models, future work could emulate the dynamic interplay between co-contraction and feedback-driven mechanisms more faithfully. This could be achieved by including a dual-layer control strategy that accounts for the independent and interactive roles of feedback and impedance modulation. The work by Calalo et al. [91] suggests that incorporating impedance control into computational models could provide a more accurate representation of how the sensorimotor system balances stability and variability under dynamic conditions. By explicitly modeling the relationship between co-contraction and feedback gains, future studies can better capture the adaptive strategies employed by humans to maintain precision in motor tasks.

## Conclusion

Our findings show that human whole-body adaptation is hierarchically organised. At the top level, the ecological control layer determines what should be optimised during motor task learning by specifying the reward/cost function used by reinforcement learning to form motor plans to be used by lower-level control. In turn, the motor-control processes include internal model learning and feedback gain tuning mechanisms in addition to the feedback control to achieve the desired plan. As such, both motor-planning and motor-control processes contribute to adaptation but at different rates and based on different optimality criteria. The motor plan is adapted quickly through reinforcement learning mechanisms after failed motion execution. Conversely, motor control adaptation is slower, and predominantly occurs as a result of internal model adaptation, irrespective of whether the movement proves to be successful.

Based on the results of this study, we argue that the optimality concept in computational motor control should be redefined from an evolutionary viewpoint where ecological fitness plays the key role in generating movements, integrating injury risk in addition to cost functions that prioritise movement execution.

## Methods

### Participants

Twenty healthy male participants (age 23.1±2.1 (SD) years; height 1.83±0.06 (SD) m; mass 77.00±9.97 (SD) kg) participated in the study. An informed written consent form, approved by the National Medical Ethics Committee of Slovenia (No. 0120–339/2017/7), was signed by all participants prior to their participation. The experimental protocol was approved by the National Medical Ethics Committee of Slovenia (No. 0120–339/2017/7) and carried out in accordance with the Oviedo Convention.

### Experimental procedure

Participants were asked to perform a series of squat-to-stand motions (trials) as depicted in Fig 1A. Movements of the foot, shank, thigh, and torso were recorded with the motion capture system Optotrak 3D Investigator (NDI, Waterloo,

Ontario) in real time with a sampling rate of 1000 Hz. We firmly attached ten active markers on the apparent axis of rotation of the left and right fifth metatarsal joints, ankles, knees, hips and shoulders. Two additional markers were placed on the back, on the spinous processes of the 1st lumbar - L1, and 1st thoracic - T1 vertebra, to measure trunk angle.

The trial started from a squat position with the trunk angle between 0° and 10° in the forward direction with respect to the vertical axis. To ensure consistency of the starting squat positions, a screen was placed in front of the participants displaying the trunk angle and the height of the L1 marker. The trial ended when participants reached a fully extended standing position. The participants were asked to perform the squat-to-stand motions in rhythm with the metronome set to 100 bpm. To avoid fatigue, there was a 30 s pause in between each trial. Following visual inspection, we excluded 1.7 % of the data due to atypical motions exhibited by the participants. A trial was considered a success if the participants reached the standing position without lifting their feet, and a failure if the participants needed to make a corrective step to keep their postural balance. All successful and failed trials were recorded and analysed. To accommodate inter-individual variability in learning rates of individual participants, we normalised the number of perturbed trials prior to the catch trial to 84. Specifically, for a subject with N perturbed trials, we computed the normalized trial index as (trial number/ N) × 84.

Experiments were performed using a custom-made force-controlled pulling mechanism which generated perturbations in the backward direction. Perturbations were applied at the waist, which allowed for a direct perturbation of the participant's centre of mass (COM). The vertical velocity of the participant's COM was used to calculate the force of perturbation $F_{pert}$ as

$$F_{pert} = K \cdot \dot{y}_{COM} \cdot m_{sub},$$

where $\dot{y}_{COM}$ is the velocity of the participant's COM in the upward direction, $m_{sub}$ is the mass of the participant, and the parameter K is a constant with a value of $3\,s^{-1}$. The constant K was experimentally determined in a preliminary squat-to-stand experiment and was set to a value that generated a perturbation at which untrained participants needed to make a corrective step to keep their postural balance. COM was calculated by using a reduced three-dimensional biomechanical model of the human body. The model consists of five body segments (left/right shank and thigh and the trunk) with their inertial properties based on specifications from De Leva [93].

## Measures and statistical analysis

To quantify motor processes, we used four measures: Trajectory Area, Initial Trajectory Area, Smoothness and Co-Contraction. Due to the different anthropometric parameters of individual participants, and due to the fact that individual trials had slightly different starting and ending points in spite of the visual feedback and the COM target specification, the trajectories were translated to start from the common origin and y was scaled to 0.5 m. This translation was for computational convenience and did not affect the statistical results.

The Trajectory Area represented the total deviation of the COM trajectory with respect to the straight line in the sagittal plane originating from the start COM position. The Initial Trajectory Area represented the COM trajectory deviation with respect to the straight line of the first 5 % of the movement, which on average occurred within $96.4 \pm 7.0$ (SD) ms from the start of the movement and approximately corresponded to 2.5 cm in height. Area, either Trajectory Area or Initial Trajectory Area, of a given perturbed trajectory $T(x(t), y(t))$, was defined as the path integral of the distance of the trajectory points to the straight line in the sagittal plane:

$$A = \int_{y_{start}}^{y_{end}} x(t)dy,$$

where $y_{start}$ is a starting COM height in the squat position, and $y_{end}$ is 2.5 cm for the Initial Trajectory Area and COM height in the standing position for the Trajectory Area. Positive values for Trajectory Area and Initial Trajectory Area represent the

forward direction (direction against the perturbation) while negative values represent backward direction (the direction of perturbation).

Smoothness was calculated as a dimensionless squared jerk [94]

$$SM = \left( \int_{t_1}^{t_2} \dddot{x}(t)^2 dt \right) D^3 / v_{mean}^2,$$

where $D$ is the duration and $v_{mean}$ is the mean speed of the movement.

Co-Contraction of the thigh antagonistic pair of muscles was calculated as a quotient of integrated muscle activity of vastus medialis $iEMG_{vm}$ and biceps femoris $iEMG_{bf}$ [95]

$$COC = \frac{iEMG_{vm}}{iEMG_{bf}}.$$

The EMG signals were recorded at a sampling rate of 1000 Hz using surface electrodes (SX230–1000, Biometrics Ltd., Newport, UK). The recordings were performed synchronously with the motion capture system. Both EMG signals were band-pass filtered using a zero-lag, 4$^{th}$-order Butterworth filter with cut-off frequencies of 20 and 500 Hz. The signals were then full-wave rectified, and root mean square (RMS) envelopes were computed. Finally, the RMS envelopes were integrated over time ($iEMG$) to express the magnitude of muscle activity.

For all four measures we used paired t-tests to compare the total change of the values between two consecutive successful trials, and between two consecutive successful trials separated by at least one failed trial in between. For each of the two conditions, we first calculated the sum of all changes during the perturbation block for each participant and then calculated the average values across all participants. This approach was chosen to capture how failed trials uniquely influence motor adaptation by comparing changes in successful movements with and without intervening failures. Since failed motions are significantly different from successful motions, directly comparing the two would primarily reflect their inherent biomechanical differences rather than the influence of failures on motor adaptation. By focusing on changes between successful trials with and without failures in between, we isolated the specific contribution of failed trials to the adaptation process. Failed trials consisting of a forward step – opposite to perturbation – represented 1.4 % of all failures. These trials were excluded from subsequent analysis. Moreover, we also calculated an average of the total change of the measures between the first and the last trial in the perturbation block. We also used paired t-tests to test if Initial Trajectory Area and Trajectory Area changed significantly from one successful trial to the next. This was tested separately for the case of consecutive successful trials, and consecutive successful trials with at least one failed trial in between.

We determined the learning rates for the average values of Initial Trajectory Area, Trajectory Area, Smoothness, Co-Contraction, and the number of failed trials using an exponential curve

$$ae^{-\frac{x}{b}} + c$$

that was fitted to average data, and where $x$ is the trial index, and the time constant $b$ represents the learning rate. Moreover, for each participant, we identified the plateau trial at which these metrics stabilised, defined as reaching 95 % of their value on the fitted exponential curve. Statistical analyses were then conducted based on these plateau values. In Co-Contraction fitting, 3 subjects were excluded due to their noisy EMG data, and 1 subject was excluded due to the negative value of the time constant $b$. In Smoothness fitting, 3 subjects were excluded due to a linear fit with infinite learning rate, and 1 subject was excluded due to the negative value of the time constant $b$. To determine if the Trajectory Area or Initial Trajectory Area of successful trials increased during the adaptation period, we compared the average values of the first and last three trials with a paired t-test.

To compare Initial Trajectory Area and Trajectory Area between catch trial, the 1st de-adaptation trial, and last perturbed trial, we performed repeated measures ANOVA and additional post-hoc paired t-tests with Bonferroni correction. The relative reductions in the Trajectory Area and Initial Trajectory Area in the first de-adaptation trial with respect to the total change due to the preceding adaptation, was calculated as

$$RD = 1 - \frac{A_{D1} - A_B}{A_C - A_B},$$

where $A$ is either Initial Trajectory Area or Trajectory Area, and the subscripts $D1$, $C$, and $B$ denote the first de-adaptation trial, catch trial, and baseline trial, respectively. Relative reductions in Trajectory Area and Initial Trajectory Area were compared with a paired t-test.

## Computational modelling

To justify the involvement of reinforcement learning with a modular and hierarchical learning system we developed a computational model where the higher-level Ecological Control module dictates which cost function to use for ecologically valid learning (Fig 5). In lower layers, Movement Policy Learning optimises which motor plan to use for squat-to-stand action, via policy gradient reinforcement learning. The plan is executed with a feedback and feed-forward control scheme, which includes an adaptive Inverse Dynamics Model. The model learns by observing the sensorimotor data generated during action and the gains of the feedback controller that are automatically tuned based on the prediction error of the Inverse Dynamics Model. The body is modelled as a two degrees-of-freedom inverted pendulum with one translational and one rotational actuator as depicted in S3 Fig. The mass and length parameters of the model were chosen to match the parameters of the human participants.

Ecological Control is modelled as a fixed logic that controls learning for ecological fitness. It is assumed that such a mechanism might have developed through evolution [3]. The primary mode of operation of Ecological Control is by deciding which cost function to optimise at a given stage of learning. In a dangerous environment the primary goal is to perform the task through a safe policy, one that is less likely to induce danger due to internal noise or external factors. Only later can effort cost be considered as necessary. In the model, effort cost is engaged when a fixed number (sampled from N(50, 50)) of successful trials is achieved for a given run. For humans, this may be considered to correspond to attaining 'confidence' in a given task. Besides arbitrating the cost functions upon a task execution failure, Ecological Control also restores the current movement policy to a safe squat-to-stand solution and biases the movement policy sampling towards safety. A detailed operation of Ecological Control is provided in S2 Algorithm with meta parameters of the model listed in S1 Algorithm.

A policy gradient reinforcement learning approach is adopted to learn a policy distribution, from which a sampled value determines a motor plan. The plan is modelled as a desired trajectory that the COM of the body should follow. The trajectory is defined via spline interpolation by using three via points that are constrained to reduce the reinforcement learning search space, improving the learning speed. Reinforcement learning then finds parameters to realise a successful stand-up movement while minimising the additional cost given by the Ecological Control module. The implementation details of the adopted policy gradient-based reinforcement learning method can be seen in S2 Algorithm, lines 3–11.

In addition to a binary success/failure signal for an executed trial, two types of cost functions are associated with the movement which are orchestrated by Ecological Control. The first cost function measures the safety of the body, quantified in relation to the extent to which the zero-moment point [96] is further away in the opposite direction of the perturbing force. The other cost function measures the effort spent by executing the given motor plan. The composite reward function realising this switching logic can be seen in the S2 Algorithm line 10.

The Internal Dynamics Model is initialised with the inverse dynamics of the body in the unperturbed environment. Then the model is allowed to learn in the perturbed environment by using the experience obtained through execution. Namely,

the model learns the new dynamics based on the net command sent to the model of the body and the observed accelerations by modifying the unperturbed dynamics with a learnable additive term (see S3 Algorithm for the details).

The movement controller combines the feed-forward signal obtained from the Internal Dynamics Model with a feedback controller output. The feedback controller is implemented as a PD controller that has an adaptable scale factor that acts as a multiplier of the initial gain parameters. The scale parameter is linked to the prediction error of the Internal Dynamics Model through a fixed inverse relation that tunes the gains during task execution over the trials.

## Supporting information

**S1 Fig. Visualization of Trajectory Area Calculation.** Illustration of how Trajectory Area is computed as the integral of the deviation from a straight-line reference in the sagittal plane. The hatched regions represent the calculated area where regions of negative area are hatched with minus signs (–), and regions of positive area are hatched with plus signs (+). (PDF)

**S2 Fig. Trial-by-trial progression of centre of mass peak displacement.** Same as Fig 1D and 1E except that the measure is Forward Peak Displacement (A) and Backward Peak Displacement (B). (PDF)

**S3 Fig. Correlates of the shape of the COM trajectory and its changes.** (A) Point-mass model used for mathematical optimization where we investigated how safety and movement efficiency influence the optimal trajectory of motion. Using a nonlinear programming solver, we determined optimal COM trajectories for (a) maximal safety where the centre of pressure (COP) was kept as far forward as possible, to maximise the postural stability with respect to the external perturbation acting in backward direction, and (b) for maximal movement efficiency where energy of motion was minimised. (B) The results of the optimization show that the optimal trajectories differ; if postural stability is maximised, the trajectory is arched more and hence both Trajectory Area and Initial Trajectory Area are larger than in the case when movement efficiency is maximised. Safe mode is shown in blue and efficient mode is shown in orange. The perturbation force of the efficient mode is larger than that of the safe mode due to a higher vertical velocity. (PDF)

**S4 Fig. Trials at which adaptation measures reached plateaus.** Trial numbers and standard deviations of when the Number of Failed Trials, Trajectory Area, Initial Trajectory Area, Co-Contraction, and Smoothness reached their respective plateaus. (PDF)

**S5 Fig. The correlation between the simulated model and participants.** Correlation for Trajectory Areas is illustrated in the upper pane, while the lower pane shows the correlation for Initial Trajectory Areas. Each bar represents a participant. The dark and light gray segments indicate the minimum and maximum correlation coefficients across the simulated Trajectory Areas and Initial Trajectory Areas, respectively. The mean correlation coefficients, along with their standard deviations, are shown using pink bars with error lines. The numbers above the bars indicate how many of the 59 simulation runs resulted in non-significant correlations ($p < .05$). Overall, the model captures the Trajectory Area patterns of participants to a high degree, as most participants exhibit significant correlations with the majority of the model-generated Trajectory Areas. However, Participant 7 shows the weakest correspondence, with 43 out of 59 simulated Trajectory Areas (59–16 = 43) still exhibiting significant correlations. In contrast, the lower pane shows that the model is less effective at capturing Initial Trajectory Area patterns for some participants. For example, participants 1, 6, and 20 exhibit a higher number of non-significant correlations, indicating that their Initial Trajectory Area patterns are not well captured by the simulation. (PDF)

**S1 Table. Statistical results for all pairwise comparisons of when the adaptation measures reached their respective plateaus.** Comparisons with significant differences are shown in bold. They are corrected for multiple comparisons using the Benjamini-Hochberg adjustment. \*\*\* < .001, \*\* < .01, \* < .05.
(PDF)

**S1 Algorithm. Computational model parameters and their initialization.**
(PDF)

**S2 Algorithm. Computational Model implementation and simulation flow.**
(PDF)

**S3 Algorithm. Algorithm used for Inverse Dynamics Learning.**
(PDF)

## Acknowledgments

We thank J. Čamernik and M. Jamšek for their assistance in preparing the experimental setup and conducting the experiments, and all the study participants for their involvement.

## Author contributions

**Conceptualization:** Jan Babič, Tjasa Kunavar, Erhan Oztop, Mitsuo Kawato.

**Formal analysis:** Jan Babič, Tjasa Kunavar, Erhan Oztop.

**Investigation:** Jan Babič, Tjasa Kunavar.

**Methodology:** Jan Babič, Tjasa Kunavar, Erhan Oztop.

**Software:** Erhan Oztop.

**Visualization:** Jan Babič, Tjasa Kunavar, Erhan Oztop.

**Writing – original draft:** Jan Babič, Tjasa Kunavar, Erhan Oztop, Mitsuo Kawato.

**Writing – review & editing:** Jan Babič, Tjasa Kunavar, Erhan Oztop, Mitsuo Kawato.

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
