## [Decision Letter · Decision Letter 0]

9 Jan 2025

PCOMPBIOL-D-24-01419

Success-Efficient/Failure-Safe Strategy for Hierarchical Reinforcement Motor Learning

PLOS Computational Biology

Dear Dr. Babič,

Thank you for submitting your manuscript to PLOS Computational Biology. After careful consideration, we feel that it has merit but does not fully meet PLOS Computational Biology's publication criteria as it currently stands. Therefore, we invite you to submit a revised version of the manuscript that addresses the points raised during the review process.

Please submit your revised manuscript within 60 days Mar 11 2025 11:59PM. If you will need more time than this to complete your revisions, please reply to this message or contact the journal office at ploscompbiol@plos.org. Please include the following items when submitting your revised manuscript:

We look forward to receiving your revised manuscript.

Kind regards,

Shlomi Haar, PhD

Academic Editor

PLOS Computational Biology

Daniele Marinazzo

Section Editor

PLOS Computational Biology

**Journal Requirements:**

At this stage, the following Authors/Authors require contributions: Jan Babič, Tjaša Kunavar, Erhan Oztop, and Mitsuo Kawato. Please ensure that the full contributions of each author are acknowledged in the "Add/Edit/Remove Authors" section of our submission form.

5) We notice that your supplementary Figures, Tables, and information are included in the manuscript file. Please remove them and upload them with the file type 'Supporting Information'. Please ensure that each Supporting Information file has a legend listed in the manuscript after the references list.

Potential Copyright Issues:

i) Figure 1A. Please confirm whether you drew the images / clip-art within the figure panels by hand. If you did not draw the images, please provide (a) a link to the source of the images or icons and their license / terms of use; or (b) written permission from the copyright holder to publish the images or icons under our CC BY 4.0 license. Alternatively, you may replace the images with open source alternatives. See these open source resources you may use to replace images / clip-art:

7) When completing the data availability statement of the submission form, you indicated that you will make your data available on acceptance. We strongly recommend all authors decide on a data sharing plan before acceptance, as the process can be lengthy and hold up publication timelines. Please note that, though access restrictions are acceptable now, your entire data will need to be made freely accessible if your manuscript is accepted for publication. This policy applies to all data except where public deposition would breach compliance with the protocol approved by your research ethics board. If you are unable to adhere to our open data policy, please kindly revise your statement to explain your reasoning and we will seek the editor's input on an exemption. Please be assured that, once you have provided your new statement, the assessment of your exemption will not hold up the peer review process.

**Reviewers' comments:**

Reviewer's Responses to Questions

Reviewer #1: General:

Here the authors conducted a crouched to stand task, where participants were physically perturbed while attempting to stand. They analysed center of mass (COM) trajectory changes over time, including examining when the participants did make a corrective step (“fail”) or did not make a corrective step (“success”). They then modelled the movement behaviour using a hierarchical model that searches for the final movement goal using a win-stay/lose-shift algorithm, while considering feedforward and feedback (PD) control. The paper has several positive aspects, including well conducted experiments and the general idea of a hierarchal model are reasonable. However there are several larger concerns that relate to conflating co-contraction with feedback control, trial-level behaviour analyses, not considering movement velocity and base of support, and some model assumptions. Can the authors please use *continuous line numbers*, as well as responding in a point-by-point manner while directing the reviewer using line numbers to the location of all changes.

Major

1.) Throughout the paper the authors conflate cocontraction and feedback gains. Cocontraction is generally thought to influence impedance control (see works by Etienne Burdet, David Franklin, and others), which is often used to handle unexpected perturbations. This is generally thought to be a different process than feedback control (e.g., Todorov, 2002; and work by several others). While it has been shown that co-cocontraction can elicit stronger feedback responses through automatic gain scaling (Pruszynski, J. A., Kurtzer, I., Lillicrap, T. P., & Scott, S. H. (2009). Temporal evolution of “automatic gain-scaling”. Journal of neurophysiology, 102(2), 992-1003.), impedance and feedback are generally considered different mechanisms that subserve motor control (Calalo et al., 2024: Journal of Neurophysiology, 129(4), 751-766).

While the authors have a behavioural assessment of impedance control through the dependent variable co-contraction, they do not have a behavioural assessment of feedback gains (i.e., short-latency, long-latency, or visuomotor). Accordingly, throughout the paper the authors should change the language from stating the co-contraction is a proxy of feedback control to impedance control, while also providing a paragraph or two on impedance and feedback control in the discussion. Further, since co-contraction is not a proxy of feedback control, there is a current disconnect with the computational model (see next comment).

2.) Computational modelling. The current computational model uses feedback gains on the positional and derivative terms (i.e., PD control). Yet the authors also have no true assessment of feedback gains in their behaviour results, making it difficult to reconcile the human behaviour with the implementation of the model. The paper does measure muscular co-contraction that is known to modulate impedance control, but there currently is no co-contraction and impedance control implemented in the model. The authors are encouraged to try to include impedance control within their model, so there is a more direct link between the model and behaviour. Alternatively, the authors will have to acknowledge this as a major limitation in the discussion.

3.) The authors did well to perform a trial level analysis based on success and failure. Similar analyses have been done in sensorimotor adaptation, where one considers changes in behaviour following success or failure (Pekny, S. E., Izawa, J., & Shadmehr, R. (2015). Reward-dependent modulation of movement variability. Journal of Neuroscience, 35(9), 4015-4024.; Therrien, A. S., Wolpert, D. M., & Bastian, A. J. (2016). Effective reinforcement learning following cerebellar damage requires a balance between exploration and motor noise. Brain, 139(1), 101-114.; Roth, A. M., Calalo, J. A., Lokesh, R., Sullivan, S. R., Grill, S., Jeka, J. J., Carter. M.J., & Cashaback, J. G. (2023). Reinforcement-based processes actively regulate motor exploration along redundant solution manifolds. Proceedings of the Royal Society B, 290(2009), 20231475.; van der Kooij, K., & Smeets, J. B. (2019). Reward-based motor adaptation can generalize across actions. Journal of Experimental Psychology: Learning, Memory, and Cognition, 45(1), 71.;). However, the authors chose specific pairings and triplets of task outcome: Success-success and success-fail-success. While there is some value in looking at all combinations for the past three trials, examining the change in behaviour following a success or failure should be sufficient.

Thus, it is suggested that the authors examine the absolute change of each dependent measure following i) a success, or ii) a failure. An example is shown below for trajectory area as follows:

a) \delta(trajectory area) = trajectory area (trial n+1) - trajectory area (n); given trial n was a success)

b) \delta(trajectory area) = trajectory area (trial n+1) - trajectory area (n); given trial n was a failure)

Please also do this for the other variables, initial trajectory angle, co-contraction, smoothness. Also, there is no need to keep the somewhat arbitrary choice of S-S and S-F-S combinations. Finally, could the authors also include peak trajectory displacement. This is useful to think of in terms of whether a stability margin / base of support might be crossed.

4.) The authors have focused on position. However, again this is somewhat disconnected from past research in the field of biomechanics where it is well known that the base-of-support and CoM velocity are also important determinants of stability. For example, does it matter how far the CoM traverses as long as it is within the base of support? The authors have considered both velocity and base of support in their model. Perhaps I am mistaken, but I do not believe the authors are able to examine base of support behaviourally given the marker placements. However, they can examine CoM velocity. Can the authors please examine CoM velocity for successes and failures, and report these data in supplementary. Further, can they also show phase space figures by plotting position and velocity against one another, and also provide this information in supplementary material. These plots could also be separate for success and failure trials.

Minor

1. The introduction is extremely broad, basically providing a brief summary of the many different subfields in sensorimotor neuroscience. However, the hypothesis revolves around the idea of the brain accounting for environmental risk / threats. Yet there is no literature review of this area, which is indeed very rich in the neuroeconomics literature including work done by Maloney and Landy, as well as some work done by Alaa Ahmed and more recently James Finley during gait. Also, the idea of fall risk also has an extremely large literature in gait and posture which have not been addressed in the introduction, including concepts such as base of support. In general, there should be a great deal of reworking/rewriting of the introduction which is currently does not sufficiently steer towards the hypothesis / question being tested.

2. When “initial trajectory area” is first introduced, please specify the points of integration. Further, looking further down the manuscript the authors use a position distance of 2.5cm. What is the time that this takes on average? Voluntary feedback control can occur as early as 100 ms. Thus, this metric to capture feedforward control should be on average at 100ms or slightly less. Can the authors please comment and adjust their distance of 2.5cm if necessary.

3. Can the authors please provide an equation when they introduce the term trajectory area. Note the equation is further below, but it is an important detail and currently unclear when reading the paper in a linear order. Further, please visually show this “area” by filling in the area being calculated with a sample trajectory, in a way that is more intuitive than currently shown in Figure 2. Intuitively one does not think of areas as negative. Perhaps just say “trajectory integral” instead of “trajectory area”. Then show a case with a trace where it is positive, and show a trace where it comes out negative. That will help the reader realize what the dependent measure means.

4. Here the authors frame a corrective step as a failure. It is often part of a successful gait repertoire during walking, which can be captured by optimal control. Will be useful to mention and cite works by Art Kuo and John Jeka on the role of step placement and stability in the discussion.

5. There has been a number of models in sensorimotor adaptation by Wolpert/Bastian and Cashaback on using reward/reinforcement to update motor plans, as well as the influence of movement variability following success and failure by Reza Shadmehr and Jeroen Smeets. Can the authors discuss similarities and differences of their approach to these classes of models and observed behaviours in the literature.

6. please change the word “don’t” to “do not”.

7. There are several typos throughout the paper. Unfortunately, it is not possible to specify where they are because there are no line numbers. Can the authors please carefully reread this manuscript and correct each of the typographical errors.

8. There are run-on sentences are throughout the manuscript. Please fix, which is readily accomplished by having one idea per sentence.

9. Define “performance error”, “internal model” when first introduced.

10. “normalised” suggests dividing by some value. I assume the authors meant they considered 84 trials before the catch trials? Also, is it not important to consider all the trials and not remove earlier trials where most of the learning takes place? Can the authors please comment.

11. When first bringing up that the model adapted the movement plan, can you please specify exactly how this was implemented.

12. Can the authors please report in the supplementary the results of the co-contraction fitting when the 3 subjects that were removed were also included.

13. Model. optimal feedback control argues against trajectory planning, but rather a more flexible response using feedback gains towards the final goal/target (see Cluff, T., & Scott, S. H. (2015). Apparent and actual trajectory control depend on the behavioral context in upper limb motor tasks. Journal of Neuroscience, 35(36), 12465-12476.). Can the authors please comment.

14. Fig. 6. Can the authors use other colours than red / green, which is difficult to see for those that are colour blind.

15. Can the authors please look at the mean power frequency of the vastus medialis to see if it decreases over the course of the experiment, which will provide an indication on whether the participant was fatigued.

Sincerely,

Joshua Cashaback

Reviewer #2: This paper integrates ecological factors (success vs. catastrophic failure) with a typical adaptation framework in which an established controller adjusts to an external perturbation. In a novel paradigm, participants here adapt to a force-field while performing squat-to-stand movements. In contrast to more typical adaptation tasks which involve arm reaching where failure is not catastrophic (participants may merely miss the reaching target), in this paradigm failure entails the ecologically-relevant danger of falling. Participants adapt to this force-field using a combination of both planning (which seems to be explicitly controlled) and implicit adaptation (which results in aftereffects even when participants are aware of the removal of the perturbation). Following failure, participants shift both their strategy and also show more overall learning. In addition, they tend to increase co-contraction. The paper presents a computational model which captures these effects.

This is a very interesting paper, with a well-designed behavioral experiment which is a welcome and ecologically-relevant addition to the repertoire of motor tasks used in the field. The modeling captures the behavioral findings well, and offers a framework to study the interaction between success/failure and sensorimotor learning, useful in generating further testable hypotheses. I have only a few questions before recommending acceptance.

1) Figure 4C/D compare changes in smoothness and co-contraction. Something that limits the comparisons is that, failures are more likely early, than late in learning, when the participants haven’t reached asymptote yet. This could mean that S-F-S is more likely early in learning compared to S-S; and thus changes between the second and first S in S-F-S could represent both learning (because we are still far from asymptote) and response to the failure itself. It might be helpful to look at these analyses during asymptote learning, though I wouldn’t expect a substantial difference.

2) Trials where adaptation was low / error was large are the ones more likely to be Failure trials. And, of course, participants respond more strongly to larger errors than smaller ones. Is there an effect of Failure (whether the participant had to perform a corrective movement) beyond what expected from the magnitude of the error during the Failure trial? One way to look at this might be, in the comparisons in Figs. 2BC/4CD, ask how does S-F-S compare to an S-S*-S trial, where S* is a trial with a large error similar to a Failure trial but where the participant did not do a corrective movement (i.e. a trial whereby trajectory area was similar to a failure trial, but was not a failure trial itself).

3) What is the purpose of offering separate exponential fits for the learning curves for successful vs. failed trials (e.g. Fig. 3BC)? These are not separate learning curves or even separate components of learning; successful and failed trials are generated by the same learning curve. Is it that failed trials reveal a different component of learning? I’m not arguing against showing these trials separately as the authors do in Figure 1D or 3BC: it is interesting to show that even failed trials show substantial adaptation.

4) I do not understand why, in Figure 4C/D,

“For each of the two conditions, we first calculated the sum of all changes during the perturbation block for each participant and then calculated the average values across all participants.”

Why not estimate these changes as average per trial change for each participant (so the units would be the same as in 4A/B) and then calculate the average values across all participants? If taking the sum for each participant, then this will depend on how many S-F-S vs. S-S pairs are present for each participant. So somebody with many S-F-S pairs but a mediocre response per pair would appear to have a large response, whereas someone with few S-F-S pairs but a strong response per pair would appear to have a small response. Please explain/clarify if my understanding is incorrect.

5) Minor: it may be helpful to give an idea of the timing of these movements. For example, initial reach area is calculated at 2.5cm into movement – how much time has elapsed since movement onset? What is the total movement time? Figure 4E gives some idea of how long a movement is but, even there, the end of movement is not denoted.

6) Minor: In Figure 4, It would be interesting to also show baseline co-contraction/smoothness as a reference.

Reviewer #3: The study by Jan Babic and colleagues introduces a novel computational motor control framework that integrates ecological fitness by alternating between success-driven efficiency and failure-driven safety, inspired by win-stay/lose-shift strategies. Through using an experimental paradigm on squat-to-stand movements under force perturbations to mimic an environmental threat, they demonstrate that humans adapt via hierarchical reinforcement learning mechanisms. They posited that humans effectively balance risk management and movement efficiency in real-world environments. Overall, this is an interesting study, particularly on the idea of integrating safety and the so-called efficiency of motor control, creating an ecologically relevant research topic.

While the manuscript is well written with good English, the authors must clear some questions before the manuscript can be accepted.

1. The authors employ some terminology across the manuscript. While the word safety seems self-explanatory, others may not be. For example, what do the authors mean by ‘efficiency’? In the later part of the manuscript, this was used together with the word energy efficiency after a series of successful movement productions.

2. The authors proposed a model using an inverted pendulum and hierarchical reinforcement learning, but the model performance is missing. The authors should also be more explicit in quantifying the goodness of fit of their model to the behavioural (empirical) data, e.g. using R-squared or AIC. Alternatively, the authors can present alternative model comparisons, which are also missing.

3. Ecological fitness: How is such framework proposed by the authors different from the exploration-exploitation behaviour in a pure reinforcement learning (reward-based) model of motor learning? Also, if the authors claim that failure is related to the safety mode. But why did the authors not compare S-S with F-F, instead of S-F-S? Didn't such F-F sequence provide a cleaner comparison with S-S?

Minor:

1. What is the function of the force plate (Figure 1A)? Any reason the authors did not incorporate any data from the Force Plate in the model?

2. If a commercial product, please include the EMG sensor product/model name, together with the sampling rate. Also, this is supposed to be in sync with the Optotrak mocap.

**Have the authors made all data and (if applicable) computational code underlying the findings in their manuscript fully available?**

Reviewer #1: Yes

Reviewer #2: **No: ** Currently, the paper states that data and code are available upon request.

Reviewer #3: Yes

PLOS authors have the option to publish the peer review history of their article (what does this mean? ). If published, this will include your full peer review and any attached files.

**Do you want your identity to be public for this peer review?** For information about this choice, including consent withdrawal, please see our Privacy Policy .

Reviewer #1: No

Reviewer #2: No

Reviewer #3: No

**Figure resubmission:**
---

## [Decision Letter · Decision Letter 1]

23 Apr 2025

Dear Dr. Babič,

We are pleased to inform you that your manuscript 'Success-Efficient/Failure-Safe Strategy for Hierarchical Reinforcement Motor Learning' has been provisionally accepted for publication in PLOS Computational Biology.

Best regards,

Shlomi Haar, PhD

Academic Editor

PLOS Computational Biology

Daniele Marinazzo

Section Editor

PLOS Computational Biology

Reviewer's Responses to Questions

**Comments to the Authors:**

Reviewer #1: The authors have done an excellent job addressing comments. Congratulations on a nice paper.

Sincerely,

Josh Cashaback

Reviewer #2: Thank you for responding to my questions and for performing the secondary analyses requested. I have no further comments.

Reviewer #3: The authors have sufficiently addressed the concerns.

**Have the authors made all data and (if applicable) computational code underlying the findings in their manuscript fully available?**

Reviewer #1: Yes

Reviewer #2: Yes

Reviewer #3: Yes

PLOS authors have the option to publish the peer review history of their article (what does this mean? ). If published, this will include your full peer review and any attached files.

**Do you want your identity to be public for this peer review?** For information about this choice, including consent withdrawal, please see our Privacy Policy .

Reviewer #1: **Yes: ** Joshua G. A. Cashaback

Reviewer #2: No

Reviewer #3: No

---

## [Editor Report · Acceptance letter]

PCOMPBIOL-D-24-01419R1

Success-Efficient/Failure-Safe Strategy for Hierarchical Reinforcement Motor Learning

Dear Dr Babič,

I am pleased to inform you that your manuscript has been formally accepted for publication in PLOS Computational Biology. Your manuscript is now with our production department and you will be notified of the publication date in due course.

With kind regards,

Anita Estes
